# Antibiotic Resistance and Genetic Variability of *Acinetobacter* spp. from Wastewater Treatment Plant in Kokšov-Bakša (Košice, Slovakia)

**DOI:** 10.3390/microorganisms11040840

**Published:** 2023-03-25

**Authors:** Jana Kisková, Adam Juhás, Soňa Galušková, Lenka Maliničová, Mariana Kolesárová, Mária Piknová, Peter Pristaš

**Affiliations:** Department of Microbiology, Institute of Biology and Ecology, Faculty of Science, Pavol Jozef Šafárik University in Košice, Šrobárova 2, 04154 Košice, Slovakia; jana.kiskova@upjs.sk (J.K.);

**Keywords:** *Acinetobacter* spp., wastewater treatment plant, antibiotic resistance, genetic variability

## Abstract

This study investigated the genetic variability and antibiotic resistance of *Acinetobacter* community depending on the stage of wastewater treatment in Kokšov-Bakša for the city of Košice (Slovakia). After cultivation, bacterial isolates were identified by matrix-assisted laser desorption/ionization time-of-flight mass spectrometry (MALDI-TOF MS), and their sensitivity to ampicillin, kanamycin, tetracycline, chloramphenicol and ciprofloxacin was examined. *Acinetobacter* spp. and *Aeromonas* spp. dominated bacterial populations in all wastewater samples. We identified 12 different groups based on protein profiling, 14 genotypes by amplified ribosomal DNA restriction analysis and 11 *Acinetobacter* species using 16S rDNA sequence analysis within *Acinetobacter* community, which showed significant variability in their spatial distribution. While *Acinetobacter* population structure changed during the wastewater treatment, the prevalence of antibiotic-resistant strains did not significantly vary depending on the stage of wastewater treatment. The study highlights the role of a highly genetically diverse *Acinetobacter* community surviving in wastewater treatment plants as an important environmental reservoir assisting in the further dissemination of antibiotic resistance in aquatic systems.

## 1. Introduction

Members of the genus *Acinetobacter* are typical gram-negative, non-motile bacteria. These bacteria were first isolated from soil by Beijerinck in 1911 and named as a species *Micrococcus calcoaceticus* [1,2]. Since then, the bacterium has been reclassified several times and was first described as the genus *Acinetobacter* by Brisou and Prevot in 1954 [3]. Currently, the genus is classified in the family Moraxellaceae, order Pseudomonadales, class Gammaproteobacteria and the phylum Proteobacteria. To date, at least 82 species have been identified, and new species are still being described [4].

Bacteria are characterized by diverse properties and can be found in a variety of environments. They are commonly found in soil, in water, on human skin and in human and animal intestinal tracts and feces. In addition, some bacteria are able to thrive in extreme environments such as raw sewage and activated sludge in wastewater treatment plants (WWTPs) or environments contaminated by heavy metals [5,6]. Some species are known to cause severe nosocomial infections, such as *A. baumannii*, *A. lwoffii* or *A. haemolyticus*. Many strains are characterized by extensive drug resistance, and some members of the genus play an important role in heavy metal removal from contaminated soils or possess hydrocarbon-degrading capabilities. Therefore, these bacteria are also interesting for application in bioremediation processes [7,8,9].

The major problem with *Acinetobacter* spp. is their resistance to many antibiotics, the most common to ampicillin, cephalothin, carbenicillin, gentamicin, amikacin, chloramphenicol, tetracycline, co-trimoxazole, ciprofloxacin, cefoperazone, etc. [10,11,12]. Mechanisms of antimicrobial resistance among *Acinetobacter* spp. depend on the species, the type of antibiotic and geographical location and can include the inactivation of antibiotics via modifying enzymes, decreased permeability of the outer membrane to antibiotics, increased efflux of the antibiotics from the cell and target site mutations [11,13]. Antimicrobial resistance is easily transferred among *Acinetobacter* spp. by plasmids and transposons, mainly via horizontal gene transfer, such as conjugation and transformation. However, gene transfer by bacteriophage transduction has also been observed among these bacteria [3,14].

Many studies have documented that WWTPs belong to the main anthropogenic sources of antibiotics, antibiotic-resistance genes and antibiotic-resistant bacteria due to high loads of hospital and municipal wastewater and insufficient degradation of antibiotics [6,15]. *Acinetobacter* spp. are commonly found in WWTPs and represent an important component of cultivable microbiota of wastewater and activated sludge. They play an important role in nitrification–denitrification processes and phosphorus removal from wastewater [16,17].

The aim of this study was to assess the genetic variability and antibiotic resistance of *Acinetobacter* spp. obtained from the wastewater treatment plant in Kokšov-Bakša near Košice city (Slovakia).

## 2. Materials and Methods

### 2.1. Sampling

WWTP near the village Kokšov-Bakša treats sewage from the city Košice (eastern Slovakia) and is located on the right bank of the Hornád River (48°39′29.9″ N 21°18′13.8″ E) (Appendix A). The average capacity of the WWTP is 950 L·s^−1,^ according to the provider Eastern Slovakia Water Utility Services Company in Košice. Wastewater sampling was conducted in April 2018 from four sampling points: 1. at the water inflow, 2. from sedimentation tank after mechanical treatment, 3. during biological treatment and 4. at the water outflow. One sample was collected from each sampling site into sterile 50 mL tube with sterile pipette. The samples were immediately placed in transfer box and transported to the laboratory for further processing.

The physicochemical characteristics of the wastewater at the time of sampling (including water temperature up to 12 °C) were provided by the Eastern Slovakia Water Utility Services Company in Košice based on regular water monitoring (Appendix A).

### 2.2. Bacteria Isolation and Cultivation

Water samples were serially diluted in sterile phosphate-buffered saline (Sigma-Aldrich, St. Louis, MO, USA), and 100 μL aliquot from each dilution was inoculated onto R2A medium (Sigma-Aldrich, USA). Samples were incubated for 48 h at the laboratory temperature range of 20–25 °C. Numbers of cultivable heterotrophic bacteria in water samples were determined as a number of colony-forming units per 1 mL of water (CFU per mL). One hundred isolates, randomly selected from each sample site, were again subcultured on R2A medium and subsequently identified using matrix-assisted laser desorption/ionization-time of flight mass spectrometry (MALDI-TOF MS) and subjected to antibiotic susceptibility testing.

### 2.3. MALDI-TOF MS Identification

Bacterial protein extracts were prepared according to the manufacturer’s instructions (Bruker Daltonics GmbH, Bremen, Germany) using 70% formic acid and acetonitrile as extract solutions. Subsequently, 1 μL of each extract was placed on a stainless steel plate, dried and then 1 μL of α-cyano-4-hydroxycinnamic acid matrix was added. Samples were analyzed by Microflex LT MALDI-TOF MS system with FlexControl v.3.0 (Bruker Daltonics GmbH, Bremen, Germany). Protein profiles were analyzed by Biotyper v.3.0 against the reference library of the integrated database v.3.3.1.0. The identification criteria by MALDI-TOF MS were as follows: a score of ≥2.0 was considered as an accurate species-level identification, a score from 1.7 to 2.0 was considered as a probable genus-level identification and a score of <1.7 was considered as not reliable identification (NRI). Main spectra library (MSP) dendrograms, generated using Biotyper v.3.0, were used to infer the relationships between isolates.

### 2.4. Antimicrobial Susceptibility Testing

Agar-dilution susceptibility testing was used to examine the antibiotic resistance of isolates. Isolates were inoculated on the Mueller–Hinton agar (Merck, KgaA, Darmstadt, Germany) supplemented with one of the following antibiotics to a final concentration: ampicillin (8 mg L^−1^), tetracycline (4 mg L^−1^), kanamycin (16 mg L^−1^), chloramphenicol (8 mg L^−1^) or ciprofloxacin (*Acinetobacter* spp. 1 mg L^−1^, other species 0.5 mg L^−1^), and cultivated overnight at the laboratory temperature. The concentrations of antibiotics were chosen as a consensus of recommendations (for *Acinetobacter* spp., *Enterobacteriaceae* and other non-*Enterobacteriaceae*) according to the criteria of European Committee on Antimicrobial Susceptibility Testing and Performance Standards for Antimicrobial Susceptibility Testing as official standards for susceptibility testing of antimicrobials [18,19].

### 2.5. Genetic Variability Determination of Acinetobacter spp.

The genetic variability of *Acinetobacter* spp. was assessed by a combination of protein profiling and amplified ribosomal DNA restriction analysis (ARDRA) [20]. Firstly, MALDI dendrogram was constructed based on protein spectra of all isolates identified as *Acinetobacter* species. Subsequently, 16S rRNA gene amplification of all isolates was performed and PCR products were digested with selected restriction enzymes. Thereafter, isolates were divided into separate groups based on ARDRA results and their position in the MSP dendrogram. Finally, representative isolates having unique ARDRA profiles were selected from each MALDI group for 16S rDNA sequence analysis.

### 2.6. Amplification of 16S rRNA Gene

A small amount of bacterial colony was suspended in 500 μL of distilled water, boiled for 10 min, agitated thoroughly and centrifuged briefly. Amplification of 16S rRNA gene was performed by polymerase chain reaction (PCR) using Taq Core Kit (Jena Bioscience, Jena, Germany) and Mastercycler^®^ Pro S (Eppendorf, Hamburg, Germany). A 50 μL of PCR mixture contained 1× Crystal Buffer, 200 μM of dNTP, 1 μM of each primer (fD1 5′-AGAGTTTGATCCTGGCTCAG-3′, rP2 5′-ACGGCTACCTTGTTACGACTT-3′), 1.25 U of Taq polymerase and 1 μL of bacterial suspension [21]. Initial denaturation at 95 °C for 5 min was followed by 35 cycles of denaturation at 95 °C for 1 min, primer annealing at 54 °C for 1 min and extension at 72 °C for 1.5 min with the final extension step at 72 °C for 5 min. PCR products were inspected for correct size (~1500 bp) on an agarose gel (1%, wt/vol) electrophoresis for 1 h at 95 V.

### 2.7. Amplified Ribosomal DNA Restriction Analysis

Amplified 16S rDNA fragments (5 μL) were separately digested for 1 h at 37 °C in 25 μL reaction mixture containing 2.5 μL of 10× NEBuffer and 5 U of restriction endonuclease as AluI, MspI or RsaI (New England Biolabs, Inc., Ipswich, MA, USA). Restriction fragments were separated by electrophoresis on 1.5% agarose gel for 1 h at 80 V. Subsequently, a specific genotype of each isolate was determined based on the combination of three restriction patterns.

### 2.8. Identification Using 16S rDNA Sequence Analysis

PCR amplicons were purified by ethanol precipitation, and concentration of DNA was measured using NanodropTM One (Thermo Scientific, Waltham, MA, USA). In both directions, 16S rDNA sequences were sequenced by Sanger sequencing method at SEQme s.r.o sequencing service (Dobříš, Czech Republic). Obtained sequences were assembled using the CAP3 tool [22] and aligned using a BLAST search tool against the GenBank 16S rDNA sequence database (http://www.ncbi.nlm.nih.gov/blast accessed on 20 September 2022). The sequences were submitted to the GenBank database and are available under accession numbers from OK235605 to OK235620.

### 2.9. Data Analysis

Phylogenetic placement of isolates was confirmed by multiple sequence comparisons against the existing 16S rRNA gene sequences of closest relatives obtained from GenBank database. Sequences were aligned using the ClustalW algorithm implemented in MEGAX v.10.2.4 [23]. Phylogenetic tree was constructed using the neighbor-joining method with 1000 bootstrap replications. The evolutionary distances were computed using Kimura 2-parameter model [24].

Past v.3.20 software was used for statistical evaluation of the results [25]. Differences in bacterial counts between sampling sites were analyzed by one-way ANOVA. Changes in the cultivable bacterial composition, in the occurrence of resistant strains and in the genetic variability in the genus *Acinetobacter* during the wastewater treatment were assessed by Chi-square test. Correlation analysis was used to evaluate the multi-resistance in *Acinetobacter* spp. Alpha diversity of bacterial populations, particularly of the genus *Acinetobacter*, was determined by Simpson’s and Shannon’s diversity indices.

The genetic composition of the *Acinetobacter* population depending on the stage of wastewater treatment was visualized by heatmap constructed using ClustVis web tool (https://biit.cs.ut.ee/clustvis/ accessed on 15 March 2022) [26].

## 3. Results

### 3.1. Bacteria Isolation and Cultivation

The average number of cultivable heterotrophic bacteria in inflow was calculated as 353,000 CFU per mL, which dropped significantly after the wastewater treatment (one-way ANOVA, *p* < 0.05) (Table 1).

### 3.2. Identification of Isolates Using MALDI-TOF MS

A total of 400 isolates (100 from each sampling site) were identified using MALDI-TOF MS. Ninety-four isolates were successfully identified with a score higher than 2.0 and 285 isolates with a score ranging from 1.7 to 2.0. These isolates represented 16 different bacterial genera belonging to three bacterial phyla such as Actinobacteria, Firmicutes and Proteobacteria (Figure 1). Twenty-one isolates showed a score of <1.7.

At the genus level, *Acinetobacter* spp. (95 isolates) and *Aeromonas* spp. (228 isolates) dominated cultivable bacterial populations in all water samples. The relative abundance of *Aeromonas* spp. decreased significantly (from 65% to 36%), while the occurrence of *Acinetobacter* spp. increased (from 18% to 35%) during the wastewater treatment (Chi-square test, *p* < 0.05). In addition, the counts of Enterobacteriaceae, as well as their species richness, significantly decreased (from 7 to 3) in outflow compared to the inflow wastewater sample.

A score of <1.7 was recorded for 21 isolates (NRI). These isolates were classified into 13 different MALDI groups, and according to their position in the MALDI dendrograms, they represented with a high probability 13 different genera belonging to Proteobacteria (18 isolates) and Firmicutes (3 isolates). Two unique NRI MALDI groups were found in the water sample from the sedimentation tank after mechanical treatment, five NRI MALDI groups (three of them were unique) were found in the sample during biological treatment and eight NRI MALDI groups (of which six were unique) in outflow water.

In addition, Simpson’s and Shannon’s indices indicated an increased alpha diversity of the cultivable bacterial community in the effluent compared to the inflow (Table 1). Based on these findings, we can assume that the composition of cultivable heterotrophic bacteria significantly changed during the wastewater treatment.

### 3.3. Antimicrobial Susceptibility Testing

Almost all bacterial isolates showed resistance to ampicillin (the occurrence ranged from 96% to 100%, depending on the sampling site). Moreover, kanamycin-resistant isolates achieved a very high prevalence (ranging from 55% to 81%). Resistance to other antibiotics occurred with significantly lower frequency (Figure 2a). No significant changes were observed in the prevalence of antibiotic-resistant isolates during the wastewater treatment process (Chi-square test, *p* < 0.05).

More than 98% of all *Acinetobacter* spp. were resistant to ampicillin (from 95% to 100% isolates, depending on the sampling site). The relative abundance of isolates resistant to other antibiotics ranged from 13% to 44%. No statistically significant differences were detected in the occurrence of antibiotic-resistant isolates depending on the stage of wastewater treatment (Chi-square test, *p* < 0.05) (Figure 2b).

Only one isolate of *Acinetobacter* spp. was sensitive to all tested antibiotics (the intrinsic resistance to ampicillin was not considered). The abundance of isolates resistant to two different antibiotics increased in affluent compared to inflow water. On the other side, antibiotic resistance to three and four antibiotics decreased during the wastewater treatment. However, the differences were not statistically significant (Chi-square test, *p* > 0.05) (Table 2). Low correlation coefficient values (between −0.03 and 0.21) indicate a weak relationship in the prevalence of antibiotic resistance among *Acinetobacter* spp. (Appendix A).

### 3.4. Identification and Genetic Variability Determination of Acinetobacter spp.

Members of the *Acinetobacter* genus were classified into 12 different MALDI groups based on their position in the MSP dendrogram using distance unit 200 as a group delineation level (Appendix A). An exception is MALDI group IV (later identified as *A. johnsonii*), including seven separate clusters at the distance level 200, which was considered as a single group based on following ARDRA and 16S rDNA sequence analyses.

We identified 14 different genotypes within *Acinetobacter* spp. based on a combination of MALDI-TOF MS protein spectra analysis and ARDRA (Appendix A). The number of different genotypes increased from five to eight during the wastewater treatment, and each sampling site was characterized by the presence of unique genotypes not found in any other water sample (Table 3).

One unique genotype (G6) of *Acinetobacter* spp. was detected in an inflow water sample, two unique genotypes (G2 and G9) were found in the population from the sedimentation tank after mechanical treatment, two genotypes (G7 and G12) were unique for population obtained during biological wastewater treatment and the bacterial community from outflow showed four unique genotypes (G1, G8, G10 and G13) (Figure 3).

A clear correlation between MALDI and ARDRA groups was observed in several *Acinetobacter* clusters, e.g., all isolates classified into MALDI groups IX and XII possessed an identical ARDRA profile (Table 3). However, at least two different ARDRA profiles were detected in MALDI groups III, IV and V. MALDI group IV included isolates belonging to three different ARDRA profiles, indicating some probable degree of genetic non-homogeneity within MALDI groups.

A total of 16 representative isolates were identified based on the 16S rDNA sequence. The analysis confirmed significant differences in the spatial distribution of *Acinetobacter* species among the stages of the wastewater treatment process (Chi-square test, *p* < 0.05). While *A. johnsonii* (genotype G3, G4 and G5) dominated the bacterial communities in the first three sampling sites, *A. movanagherensis* Movanagher 4 (genotype G14) dominated the bacterial population in the effluent water. The 16S rDNA sequence analyses showed the highest similarity of the genotype G8 to *A. bouvetii* DSM 14964 (99.12%), G11 to *A. johnsonii* ATCC1 7909 (98.23%) and G12 to *A. kyonggiensis* KSL5401-037 (98.05%) (Table 3). However, subsequent phylogenetic analysis showed that these three genotypes might represent, with a high probability, new species belonging to the genus *Acinetobacter*. These findings are supported by branch lengths and low bootstrap values in the phylogenetic tree (Figure 4).

All analyses (MALDI-TOF MS, ARDRA, 16S rDNA sequence) indicated the high variability of *Acinetobacter* spp. community, which increased during the wastewater treatment process (Table 4). Generally, Simpson’s and Shannon’s diversity indices indicated the highest diversity of *Acinetobacter* spp. communities evaluated based on different protein spectra.

Interestingly, based on 16S rRNA gene analysis, MALDI groups III and IV with distance levels over 900 belong to the same *A. johnsonii* species (Appendix A). In addition, ARDRA indicated the presence of the same genotypes in both MALDI groups (G3 and G4). Nevertheless, the groups differed significantly in their protein spectra, and a relatively high variability of protein spectra was also found within MALDI group IV alone.

## 4. Discussion

Antimicrobial resistance is a natural phenomenon that occurs via natural selection, but anthropogenic activities such as the use of antibiotics for therapeutic purposes lead to a significant increase in antibiotic resistance in various components of the environment [27,28,29].

The efficiency of WWTPs in the reduction of pathogenic microorganisms and antibiotic degradation varies depending on the purification process. Some studies documented that the number of bacteria, including resistant strains, was reduced up to 97–99% during the wastewater treatment process, but the degradation of antibiotics remained insufficient in many cases [30,31,32,33]. A high microbial density in WWTP (mainly in activated sludge) and the presence of antibiotics at sub-inhibitory concentrations may facilitate a horizontal gene transfer among bacterial communities. Therefore, WWTPs can be important reservoirs of antibiotic-resistant bacteria and antibiotic-resistance genes persisting in the final effluent and spreading into the environment [6,34].

A recent study demonstrated that the wastewater treatment process in Kokšov-Bakša reduced more than 98.9% of cultivable heterotrophic bacteria in the effluent compared to the inflow water. Thus, we can confirm that the wastewater treatment plant has the desired efficiency in reducing the number of potentially dangerous bacteria entering the environment from wastewater. However, some metagenomic approaches demonstrated that the whole bacterial community was not significantly reduced by the purification process, but only its species composition changed. Generally, fecal bacteria and other potential pathogens (e.g., *Escherichia, Shigella* species) are dramatically reduced after water treatment [6,35]. Similarly, there was a significant reduction in bacteria of the family Enterobacteriaceae as well as of the genus *Aeromonas* after the purification process in a recent study.

Almost all isolates showed resistance to ampicillin. This result could be expected, as ampicillin has long been one of the most widely used antibiotics in Slovakia [36]. In addition, the abundance of antibiotic-resistant strains remained approximately at the same level in different stages of wastewater treatment despite the changing composition of the cultivable bacterial population. We can assume that this phenomenon is caused by residual antibiotics and a horizontal gene transfer between bacteria.

Generally, *Acinetobacter* spp. are characterized by a natural resistance to penicillins, so a high number of ampicillin-resistant strains in wastewater could be expected [37]. In addition, *Acinetobacter* spp. show multi-resistance to many antibiotics, including various beta-lactams [10,11,12]. Similarly, in WWTP of Kokšov-Bakša, 18.94% of isolates showed resistance to two antibiotics, 4.21% to three and 2.11% to four different antibiotics, whereas the number of multiresistant isolates did not change significantly during the wastewater purification process. *Acinetobacter* strains resistant to ampicillin, kanamycin, erythromycin, tetracycline and chloramphenicol were also detected in the effluent in a previous study [38]. Hubeny et al. [39] demonstrated that the number of *Acinetobacter* isolates carrying carbapenem antibiotic resistance genes increased in river water samples collected downstream from the wastewater discharge point (18.6% of isolates) compared to water samples collected upstream from the wastewater discharge point (13.2% of isolates).

A single-medium culture method may provide very limited results on the structure of the bacterial population in the wastewater treatment plant. However, several previous culture-dependent and especially culture-independent (metagenomic) approaches have shown that *Acinetobacter* spp. are an important part of the microbial community of WWTPs. Many species show a high ability to survive the water purification process [6,40,41,42]. Some species, such as *A. junii* and *A. venetians*, are even able to survive water chlorination [41].

The data on the species composition of *Acinetobacter* populations in WWTPs are relatively scarce, as most previous studies reported the presence of *Acinetobacter* spp. in wastewater at the genus level with no further species identification. There are several studies focusing on clinically important species, such as *Acinetobacter baumannii,* isolated from wastewater [43,44]. The presence of *Acinetobacter johnsonii* was also confirmed in WWTPs [45]. High diversity of the genus *Acinetobacter* was reported in different WWTPs by Pulami et al. [46]. They phylogenetically identified (based on 16S rRNA gene and rpoB sequence analyses) a total of 132 *Acinetobacter* isolates classified in 14 different phylotypes.

Our study revealed remarkably high variability of *Acinetobacter* spp. community in WWTP using various methods. Based on 16S rDNA sequence analysis, at least 11 different *Acinetobacter* species were identified whose spatial distribution varied significantly depending on the stage of the purification process. While *A. johnsonii* dominated the *Acinetobacter* community in inflow water (84.9%), and its prevalence decreased during the water treatment process, *Acinetobacter movonagherensis* dominated the community in outflow water (45.71%). In fact, some stages of the purification process were characterized by species not found in other phases. In addition, at least three isolates could represent new species of the genus *Acinetobacter* spp., but further analyses are needed to evaluate these preliminary findings. Finally, we also assume the presence of *Acinetobacter baumannii* in WWTP in Kokšov-Bakša (confirmed in previous unpublished pilot experiments), which was not detected by the culture method used in a recent study.

A high genetic diversity of *Acinetobacter* community was also confirmed using ARDRA with three different genotypes detected within *A. johnsonii* and two different genotypes within *A. junii* community. Moreover, biotyping using MALDI-TOF MS revealed unusually high heterogeneity of protein profiles within *A. johnsonii* group. Similar results were obtained when only 16 out of 23 *A. johnsonii* isolates (identified by 16S rRNA gene sequencing) could be securely identified by the MALDI-TOF MS approach [47]. All these data indicate the genetic non-homogeneity of *A. johnsonii* species and the existence of several genospecies in *A. johnsonii* complex. Modern genomic data strongly suggest that the diversity of the *Acinetobacter* genus is heavily underestimated, and multiple new species have been proposed based on molecular data [48].

One study demonstrated that wastewater treatment contributes to the selective increase of antibiotic resistance among *Acinetobacter* spp. [49]. Our findings showed that despite changes in *Acinetobacter* population composition during the wastewater treatment process, the relative abundance of antibiotic-resistant isolates did not fluctuate.

## 5. Conclusions

In conclusion, our results indicate that the WWTP for Košice in Kokšov-Bakša may represent a potential source of resistant bacteria that are further spread into the aquatic environment of Hornád River. The genus *Acinetobacter* is an important component of the cultivable heterotrophic microbiota in WWTP, exhibiting a high biodiversity and species richness, which varied depending on the stage of wastewater treatment. Finally, we hypothesize that *Acinetobacter* spp. persisting permanently or for a prolonged time in various stages of wastewater plants could, through horizontal gene transfer, significantly contribute to the emergence and dissemination of new antibiotic resistance bacteria entering the environment by effluent water.

## Figures and Tables

**Figure 1 microorganisms-11-00840-f001:**
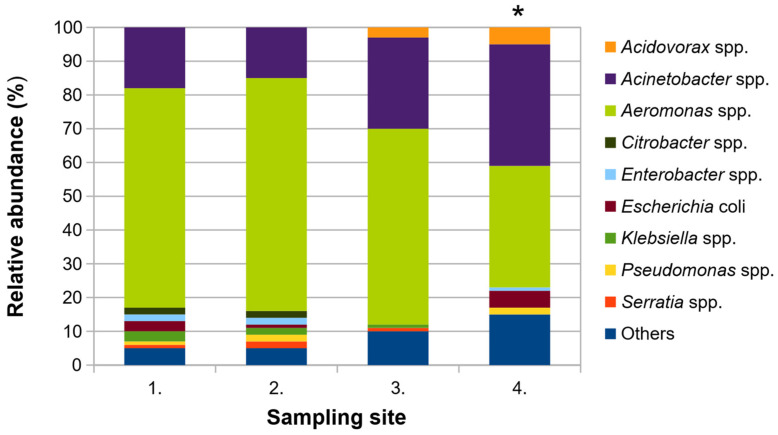
The composition of cultivable bacterial communities in different stages of the wastewater treatment process in Kokšov-Bakša, (Košice, Slovakia). The category “Others” groups bacteria whose relative abundance was within the community below 2%, such as *Achromobacter* spp., *Kluyvera* spp., *Lactobacillus* spp., *Lactococcus* spp., *Leuconostoc* spp., *Providencia* spp., *Raoultella* spp. and unreliable identified isolates by MALDI-TOF MS. 1.—inflow, 2.—mechanical wastewater treatment, 3.—biological wastewater treatment, 4.—outflow. * Significant reduction in the number of *Aeromonas* spp. and significant increase in the number of *Acinetobacter* spp. during the wastewater treatment process (Chi-square test, *p* < 0.05).

**Figure 2 microorganisms-11-00840-f002:**
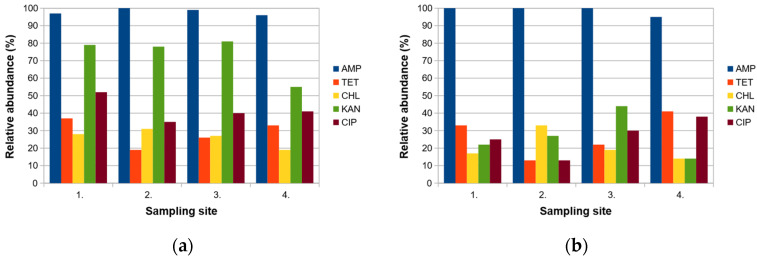
The prevalence of antibiotic-resistant strains within cultivable bacterial community (**a**) and *Acinetobacter* spp. (**b**) in different stages of the wastewater treatment process in Kokšov-Bakša (Košice, Slovakia). AMP—ampicillin, KAN—kanamycin, TET—tetracycline, CHL—chloramphenicol, CIP—ciprofloxacin. 1.—inflow, 2.—mechanical wastewater treatment, 3.—biological wastewater treatment, 4.—outflow.

**Figure 3 microorganisms-11-00840-f003:**
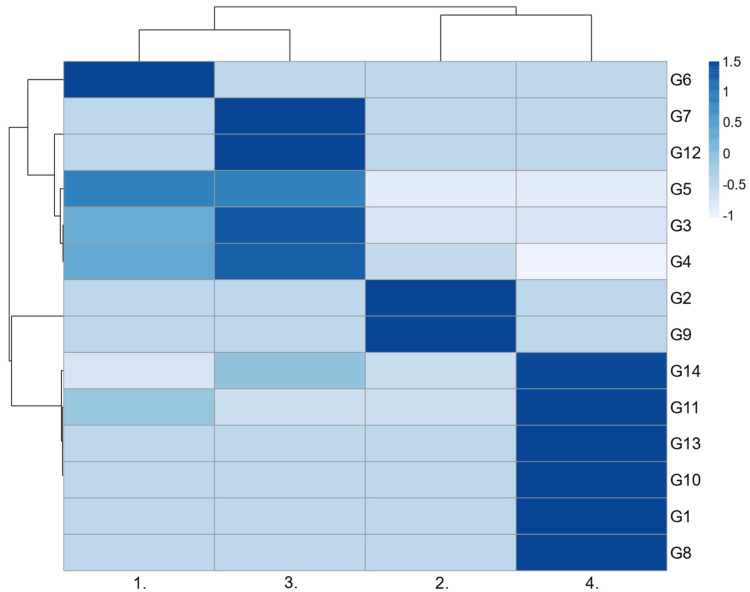
Genotype distribution of *Acinetobacter* spp. during the wastewater treatment. The darkest color shows the unique genotypes (G1–G14) of *Acinetobacter* spp. for sampling site. 1.—inflow, 2.—mechanical wastewater treatment, 3.—biological wastewater treatment, 4.—outflow.

**Figure 4 microorganisms-11-00840-f004:**
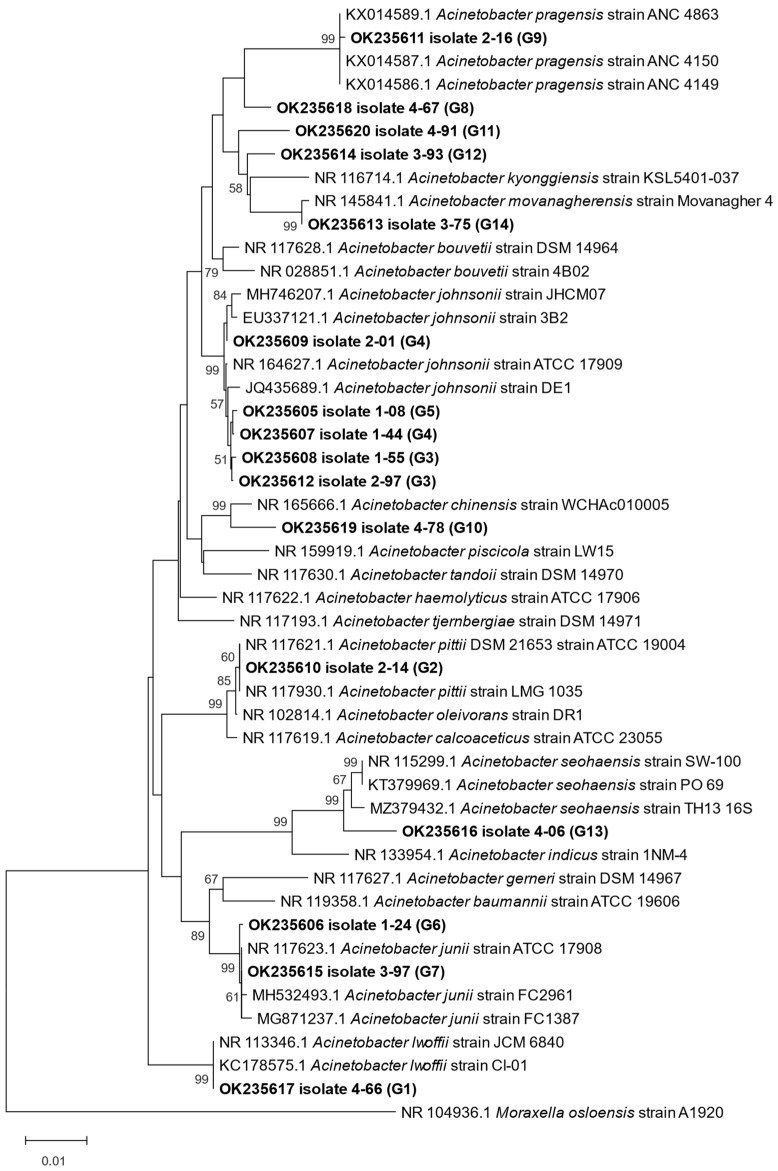
Phylogenetic placement of representative *Acinetobacter* strains (OK235605-OK235620) from the wastewater treatment plant within the genus *Acinetobacter*. G1–G14—genotypes obtained based on digestion of 16S rRNA gene using three different restriction enzymes AluI, MspI and RsaI. Phylogenetic tree was constructed using neighbor-joining method with 1000 bootstrap replications. The evolutionary distances were computed using Kimura 2-parameter model.

**Table 1 microorganisms-11-00840-t001:** Abundance and biodiversity of cultivable heterotrophic bacterial community in water samples from different stages during the wastewater treatment process in Kokšov-Bakša, Košice (Slovakia).

Sampling Site	Title CFU ^a^ Per mL Wastewater	Simpson’s Index	Shannon’s Index
1. inflow	353 × 10^3^	0.5418	1.278
2. mechanical treatment	116 × 10^4^	0.4986	1.194
3. biological treatment	61 × 10^3^	0.5716	1.226
4. outflow	3.6 × 10^3^ *	0.7372	1.800

^a^ CFU—colony forming unit; average number of CFU per mL calculated from three different dilutions of each water sample. * Significant reduction in the number of CFU per mL in outflow compared to inflow water (one-way ANOVA, *p* < 0.05).

**Table 2 microorganisms-11-00840-t002:** The prevalence of multiresistant *Acinetobacter* isolates from the wastewater treatment plant Kokšov-Bakša (Košice, Slovakia).

Number of Antibiotics	Number of Resistant Isolates (%)			
	1. (n ^a^ = 18)	2. (n = 15)	3. (n = 27)	4. (n = 35)
2	1 (5.56)	2 (13.33)	7 (25.93)	9 (25.71)
3	1 (5.56)	1 (6.67)	1 (3.70)	1 (2.86)
4	1 (5.56)	0 (0)	1 (3.70)	0 (0)

^a^ n—total number of *Acinetobacter* isolates in wastewater sample. The intrinsic resistance to ampicillin was not considered. 1.—inflow, 2.—mechanical wastewater treatment, 3.—biological wastewater treatment, 4.—outflow.

**Table 3 microorganisms-11-00840-t003:** Variability of *Acinetobacter* community in wastewater treatment plant in Kokšov-Bakša for Košice city (Slovakia) based on MALDI-TOF MS, ARDRA and 16S rDNA sequence analysis.

MALDI Group	Genotype (Representative Strain)	Accession No. (GenBank)	Restriction Enzyme			Number of Isolates in Sampling Site (%)				Best Hit Blastn Against 16S rDNA Sequence Database (Similarity)
			AluI	MspI	RsaI	1. ^a^	2.	3.	4.	
I	G1 (4-66) ^b^	OK235617	3 ^c^	3	4	0	0	0	1 (2.86)	*A. lwoffii* JCM 6840 (100%)
II	G2 (2-14)	OK235610	4	2	4	0	1 (6.67)	0	0	*A. pitii* LMG 1035 (100%)
III	G3 (1-55)	OK235608	3	1	2	1 (5.56)	0	0	1 (2.86)	*A. johnsonii* ATCC 17909 (99.92%)
	G4 (1-44)	OK235607	3	3	2	2 (11.11)	0	0	0	*A. johnsonii ATCC* 17909 *(99.93%)*
IV	G3 (2-97)	OK235612	3	1	2	2 (11.11)	2 (13.33)	4 (14.81)	1 (2.86)	*A. johnsonii* ATCC 17909 (99.85%)
	G4 (2-01)	OK235609	3	3	2	10 (55.56)	10 (66.67)	14 (51.85)	9 (25.71)	*A. johnsonii* ATCC 17909 (99.92%)
	G5 (1-08)	OK235605	1	1	1	1 (5.56)	0	1 (3.70)	0	*A. johnsonii* ATCC 17909 (99.92%)
V	G6 (1-24)	OK235606	2	2	2	1 (5.56)	0	0	0	*A. junii* ATCC 17908 (99.93%)
	G7 (3-97)	OK235615	8	2	2	0	0	2 (7.41)	0	*A. junii* ATCC 17908 (100%)
VI	G8 (4-67)	OK235618	5	3	5	0	0	0	1 (2.86)	*A. bouvetii* DSM 14964 (99.12%)
VII	G9 (2-16)	OK235611	3	3	4	0	1 (6.67)	0	0	*A. pragensis* ANC 4149 (99.71%)
VIII	G10 (4-78)	OK235619	3	2	5	0	0	0	1 (2.86)	*A. chinensis* WCHAc010005 (99.04%)
IX	G11 (4-91)	OK235620	5	3	4	1 (5.56)	0	0	4 (11.43)	*A. johnsonii* ATCC1 7909 (98.23%)
X	G12 (3-93)	OK235614	7	3	5	0	0	1 (3.7)	0	*A. kyonggiensis* KSL5401-037 (98.05%)
XI	G13 (4-06)	OK235616	3	2	4	0	0	0	1 (2.86)	*A. seohaensis* SW-100 (98.82%)
XII	G14 (3-75)	OK235613	6	3	2	0	1 (6.67)	5 (18.52)	16 (45.71)	*A. movanagherensis* Movanagher 4 (99.41%)
Total number of isolates						18	15	27	35	
Chi-square test						*p* < 0.05 ^d^				

^a^ 1.—inflow, 2.—mechanical wastewater treatment, 3.—biological wastewater treatment, 4.—outflow. ^b^ Genotype obtained using amplified ribosomal DNA restriction analysis (ARDRA). ^c^ A type of restriction pattern obtained after digestion with the appropriate restriction enzyme. ^d^ Significant differences in genotype distribution of *Acinetobacter* spp. during wastewater treatment evaluated by Chi-square test (*p* < 0.05).

**Table 4 microorganisms-11-00840-t004:** Alpha diversity of *Acinetobacter* spp. population in different stages of the wastewater treatment process observed using different method.

Diversity Index	Analysis	Sampling Site			
		1.	2.	3.	4.
Simpson’s	MALDI-TOF MS	0.5185	0.5333	0.7462	0.7069
	ARDRA	0.5185	0.5244	0.6667	0.7053
	16S rRNA gene	0.2037	0.3467	0.4636	0.6759
Shannon’s	MALDI-TOF MS	1.051	1.173	1.739	1.564
	ARDRA	1.051	1.081	1.373	1.525
	16S rRNA gene	0.426	0.720	0.874	1.376

Sampling site: 1.—inflow, 2.—mechanical wastewater treatment, 3.—biological wastewater treatment, 4.—outflow. MALDI-TOF MS—matrix-assisted laser desorption/ionization-time of flight mass spectrometry, ARDRA—amplified ribosomal DNA restriction analysis.

## Data Availability

The data supported in this study are available upon request from the corresponding author, and 16S rDNA sequences of *Acinetobacter* spp. are openly available in the GenBank database under accession numbers from OK235605 to OK235620.

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
