# Peer review of "Antibiotic Resistance and Genetic Variability of Acinetobacter spp. from Wastewater Treatment Plant in Kokšov-Bakša (Košice, Slovakia)"

_microorganisms, 2023, doi:10.3390/microorganisms11040840_

Round 1

Reviewer 1 Report

The dynamics and diversity of antibiotic resistant bacteria are important for our health. The authors showed a potential source of resistant bacteria in Kokšov-Bakša. The location is limited but the results are important for antibiotic resistant study.

My comments are described below:

Line 83:Please show the range of lab temperature. How much temperature are water sample?

Line 90: Please describe extract solution (acetonitrile and formate?).

Line 97: In my experience, some bacterial genus identified by MALDI-TOFMS (the score =1.96) is classified to a different genus by 16S rRNA gene analysis. How about the genus isolated in this study ?

Line 189:Lactobacillus group was reclassified as Lactobacillus, Lacticaseibacillus, Lactiplantibacillus etc. Were isolates identified as only Lactobacillus in this study ?

Figure 1: Multidrug-Resistant Acinetobacter presents an increasing challenge to health care (Antimicrob Agents Chemother. 2010 Dec;54(12):5316-22; J Appl Microbiol. 2021 Dec;131(6):2715-2738.). Among antibiotics, Carbapenems are a class of very effective antibiotic agents.  Other effective antibiotics should be used in this study. This make the manuscript more attractive.

Author Response

Dear Sir

thanks you for your comments and suggestions that allowed us to greatly improve the quality of the manuscript. Below please find our replies

Sincerely

Peter Pristas

Comments and Suggestions for Authors

The dynamics and diversity of antibiotic resistant bacteria are important for our health. The authors showed a potential source of resistant bacteria in Kokšov-Bakša. The location is limited but the results are important for antibiotic resistant study.

My comments are described below:

Line 83: Please show the range of lab temperature. How much temperature are water sample?

Authors: The range of lab temperature was added - line 83. The wastewater temperature at the time of sampling is showed in Supplementary material TableS1; information also added in the text of manuscript in lines 76-77.

Line 90: Please describe extract solution (acetonitrile and formate?).

Author: Information was added in the text - lines 91-92.

Line 97: In my experience, some bacterial genus identified by MALDI-TOFMS (the score =1.96) is classified to a different genus by 16S rRNA gene analysis. How about the genus isolated in this study?

Authors: Isolates identified as the members of Acinetobacter with the score of 1.7-2.0 by MALDI-TOF MS were included in further analysis. They were confirmed as Acinetobacter spp. based on 16S rRNA analysis.

Line 189: Lactobacillus group was reclassified as LactobacillusLacticaseibacillus, Lactiplantibacillus etc. Were isolates identified as only Lactobacillus in this study?

Authors: Yes, isolates were identified as only Lactobacillus spp..

Figure 1: Multidrug-Resistant Acinetobacter presents an increasing challenge to health care (Antimicrob Agents Chemother. 2010 Dec;54(12):5316-22; J Appl Microbiol. 2021 Dec;131(6):2715-2738.). Among antibiotics, Carbapenems are a class of very effective antibiotic agents.  Other effective antibiotics should be used in this study. This make the manuscript more attractive.

Authors: Yes, we agree that bacterial resistance to carbapenems is a very interesting area of research. We plan to address the issue of beta-lactamase resistance (including carbapenems) in Acinetobacter spp. in a separate study.

Reviewer 2 Report

Dear authors,

This is a interesting paper speaking about antibiotic resistance profiles and genetic variability in A. baumannii strains. However, I have some recommendations to improve this manuscript:

1. Lines 10-11 -> here is a repeated information, e.g., "in the process of wastewater treatment in the wastewater treatment plant" ; maybe you can reformulate this sentence.

2. Line 32 -> modify in italic the Moraxellaceae family.

3. Line 35 -> You could be more clear about these diverse properties of bacteria - acquisition of the resistance and virulence mechanisms and ability to survive in extreme environmental conditions. 

4. Line 39-> When you say, "others are known to cause nosocomial infections..", I understand that you speak about other bacteria apart from the ones you talked about previously that can live in extreme environments and WWTPs. However, the bacteria that cause nosocomial infections are also found in the previously mentioned environments.

5. Line 41 -> "extreme drug" can be noted as extensively-drug resistant -XDR.

6. Line 50 - here must be "decreased permeability of the..".

7. Line 108-109 -> modify in italic the name of bacteria.

8. Line 156 -> here must to be "for statistical evaluation"

9. Line 183 -> put in italic the bacteria. Please verify all the bacteria names in the manuscript.

10. Line 179 ->  You say that you identifed 100 isolates from each sampling sites =>400 isolates. 379 isolates were successfully identified with a score higher than 2.0, 21 isolates with a score <1.7. Maybe you add this numbers to be easier to identify in the text.

11.  Line 268 -Figure 3 -> I think this is more a tabel. Also, it seems that the image has been superimposed on another figure. Please check this.

12. Please add more new references, e.g.:

doi: 10.1007/s00284-022-02815-7. PMID: 35258680.

doi: 10.1016/j.jgar.2022.02.019. Epub 2022 Mar 5. PMID: 35257970.

doi: 10.1016/j.scitotenv.2022.160182. Epub 2022 Nov 14. PMID: 36395844.

doi: 10.1016/j.scitotenv.2022.153437. Epub 2022 Feb 2. PMID: 35122847.

doi: 10.1186/s13756-022-01156-1. PMID: 36104761; PMCID: PMC9476303.

Author Response

Dear Sir

thanks you for your comments and suggestions that allowed us to greatly improve the quality of the manuscript. Below please find our replies

Sincerely

Peter Pristas

Comments and Suggestions for Authors

Dear authors,

This is a interesting paper speaking about antibiotic resistance profiles and genetic variability in A. baumannii strains. However, I have some recommendations to improve this manuscript:

  1. Lines 10-11 -> here is a repeated information, e.g., "in the process of wastewater treatment in the wastewater treatment plant"; maybe you can reformulate this sentence.

Authors: Corrected.

  1. Line 32 -> modify in italic the Moraxellaceaefamily.

Authors: Only names of bacterial species and genera are written in italics according to the standard of the journal Microorganisms.

  1. Line 35 -> You could be more clear about these diverse properties of bacteria - acquisition of the resistance and virulence mechanisms and ability to survive in extreme environmental conditions. 

Authors: The sentence “Bacteria are characterized by diverse properties and can be found in a variety of environments“ is the introductory sentence of the paragraph. The next sentences separately mention various environments and various characteristics of the Acinetobacter species.

  1. Line 39-> When you say, "others are known to cause nosocomial infections.", I understand that you speak about other bacteria apart from the ones you talked about previously that can live in extreme environments and WWTPs. However, the bacteria that cause nosocomial infections are also found in the previously mentioned environments.

Authors: Corrected.

  1. Line 41 -> "extreme drug" can be noted as extensively-drug resistant -XDR.

Authors: Corrected.

  1. Line 50 - here must be "decreased permeability of the..".

Authors: Corrected.

  1. Line 108-109 -> modify in italic the name of bacteria.

Authors: Only names of bacterial species and genera are written in italics according to the standard of the journal Microorganisms.

  1. Line 156 -> here must to be "for statistical evaluation"

Authors: Corrected.

  1. Line 183 -> put in italic the bacteria. Please verify all the bacteria names in the manuscript.

Authors: Only names of bacterial species and genera are written in italics according to the standard of the journal Microorganisms.

  1. Line 179 ->  You say that you identified 100 isolates from each sampling sites =>400 isolates. 379 isolates were successfully identified with a score higher than 2.0, 21 isolates with a score <1.7. Maybe you add this numbers to be easier to identify in the text.

Authors: Total numbers of isolates identified by MALDI-TOF MS are given in lines 179-183.

  1. Line 268 -Figure 3 -> I think this is more a table. Also, it seems that the image has been superimposed on another figure. Please check this.

Authors: Figure 3 is a heatmap (output from the online tool ClustVis). It is an image not a table. Possibly there were problems during conversion from *.doc format

  1. Please add more new references, e.g.:

doi: 10.1007/s00284-022-02815-7. PMID: 35258680.

doi: 10.1016/j.jgar.2022.02.019. Epub 2022 Mar 5. PMID: 35257970.

doi: 10.1016/j.scitotenv.2022.160182. Epub 2022 Nov 14. PMID: 36395844.

doi: 10.1016/j.scitotenv.2022.153437. Epub 2022 Feb 2. PMID: 35122847.

doi: 10.1186/s13756-022-01156-1. PMID: 36104761; PMCID: PMC9476303.

Authors: The publication “doi: 10.1016/j.scitotenv.2022.153437. Epub 2022 Feb 2. PMID: 35122847” was already cited in our manuscript as reference Nr. 39. Other new references were added; Nr. 10, 12, 46.

Round 2

Reviewer 1 Report

The manuscript was improved well.